# The Induction of Long-Term Potentiation by Medial Septum Activation under Urethane Anesthesia Can Alter Gene Expression in the Hippocampus

**DOI:** 10.3390/ijms241612970

**Published:** 2023-08-19

**Authors:** Yulia V. Dobryakova, Konstantin Gerasimov, Yulia S. Spivak, Tinna Korotkova, Alena Koryagina, Angelina Deryabina, Vladimir A. Markevich, Alexey P. Bolshakov

**Affiliations:** Institute of Higher Nervous Activity and Neurophysiology, Russian Academy of Sciences, 117485 Moscow, Russiagerasimov.konstant@gmail.com (K.G.);

**Keywords:** medial septum, dorsal hippocampus, ventral hippocampus, gene expression, M1 muscarinic receptors, 192IgG-saporin

## Abstract

We studied changes in the expression of early genes in hippocampal cells in response to stimulation of the dorsal medial septal area (dMSA), leading to long-term potentiation in the hippocampus. Rats under urethane anesthesia were implanted with stimulating electrodes in the ventral hippocampal commissure and dMSA and a recording electrode in the CA1 area of the hippocampus. We found that high-frequency stimulation (HFS) of the dMSA led to the induction of long-term potentiation in the synapses formed by the ventral hippocampal commissure on the hippocampal CA1 neurons. One hour after dMSA HFS, we collected the dorsal and ventral hippocampi on both the ipsilateral (damaged by the implanted electrode) and contralateral (intact) sides and analyzed the expression of genes by qPCR. The dMSA HFS led to an increase in the expression of *bdnf* and *cyr61* in the ipsilateral hippocampi and *egr1* in the ventral contralateral hippocampus. Thus, dMSA HFS under the conditions of degeneration of the cholinergic neurons in the medial septal area prevented the described increase in gene expression. The changes in *cyr61* expression appeared to be dependent on the muscarinic M1 receptors. Our data suggest that the induction of long-term potentiation by dMSA activation enhances the expression of select early genes in the hippocampus.

## 1. Introduction

The hippocampus is one of the brain areas critically involved in memory formation. The medial septal area (MSA), which consists of the medial septum and diagonal bands of Broca’s area, provides cholinergic innervation of the hippocampus. It was shown that degeneration of these cholinergic neurons, which results in a deficit of acetylcholine in the hippocampus, may occur under some pathological conditions like Alzheimer’s disease [1], Parkinson’s disease, and dementia with Lewy bodies [2]. Modeling this deficit in animals through elimination of the cholinergic neurons in the MSA revealed learning impairments in some behavioral paradigms [3,4,5,6,7], suggesting that the acetylcholine released by septal neurons plays an important role in the processes of memory formation.

Changes in the efficacy of synaptic transmission, particularly long-term potentiation (LTP), are believed to contribute to learning and memory formation. It was shown that modulation of the acetylcholine receptors in the hippocampus may cause either potentiation or depression of glutamatergic synapses [8,9]. More specifically, it was shown that LTP in the synapses between the Schaeffer collaterals and CA1 pyramidal neurons may be induced by a high-frequency stimulation of the ventral hippocampal commissure [10], which includes both Schaeffer collaterals and axons coming from the MSA. Detailed analysis of LTP induction in this paradigm showed that LTP in the CA1 area may be induced not only by activation of the ventral hippocampal commissure but also the MSA alone [11]. LTP caused by tetanic stimulation of the MSA may be abolished by elimination of MSA cholinergic neurons, suggesting that acetylcholine released in the hippocampus during MSA stimulation may be one of the factors that promotes LTP formation and supports the processes of memory formation.

It is important to stress that in the aforementioned studies, LTP was induced by stimulation of the dorsal part of the MSA (dMSA), whose stimulation induces a clear postsynaptic response in the hippocampus. The more ventral parts of the MSA (the medial septal nucleus and diagonal bands of Broca’s area) contain more cholinergic cells than the dMSA, and their stimulation does not result in an electrophysiological response in the hippocampus [11] but can alter gene expression in the hippocampus [12]. Moreover, low-frequency dMSA stimulation had practically no effect on hippocampal cells in terms of transcriptional activity [12]. The latter is quite paradoxical since dMSA activation may induce LTP, which requires changes in the expression of genes for long-term maintenance of acquired changes in the synaptic strength [13,14]. Therefore, the aim of this study was to analyze changes in the gene expression in the hippocampus after the induction of plastic changes in the synapses formed by Schaffer collaterals on the CA1 pyramidal neurons with stimulation of the dMSA and the dependence of these changes on the cholinergic MSA neurons.

## 2. Results

### 2.1. Loss of Cholinergic Neurons in the Medial Septum Does Not Lead to Transcriptomic Changes in the Dorsal Hippocampus under Basal Conditions

To analyze the role played by septal cholinergic afferents in the regulation of gene expression in the hippocampus, we used intraseptal injection of the immunotoxin 192IgG-saporin. However, we previously showed that intracerebroventricular injection of 192IgG-saporin affects the functioning of the dorsal hippocampus [7] and some neocortical areas [15] and practically does not influence gene expression in the ventral hippocampus [7]. Therefore, our first step was to examine changes in the expression of genes in the dorsal hippocampus after induction of a cholinergic deficit by intraseptal injection of 192IgG-saporin. To this aim, we performed RNAseq of the dorsal hippocampus of control animals and animals after intraseptal injection of 192IgG-saporin. Analysis of the differential expression of genes in the hippocampus showed that the loss of cholinergic innervation did not induce substantial changes in the expression of genes with *p_adjusted_* < 0.1 (Appendix A). Principle component analysis did not reveal any component that may be characteristic of any experimental group (Appendix A). These data suggest that a cholinergic deficit in the hippocampus after intraseptal injection of 192IgG-saporin does not induce strong changes like an intracerebroventricular injection.

### 2.2. Degeneration of Cholinergic Neurons in the dMSA Prevents Expression of Some Early Genes after the Induction of Long-Term Potentiation

Our next step was to analyze changes in the gene expression that occurred after the induction of LTP in the glutamatergic synapses of the hippocampus after dMSA stimulation. In order to accomplish this, we compared three groups of animals: (1) control animals that received rhythmic low-frequency stimulation of the ventral hippocampal commissure (VHC) (“control”), (2) animals that received rhythmic low-frequency VHC stimulation and in which LTP was induced by dMSA stimulation (“LTP” group), and (3) animals with cholinergic deficits that received rhythmic low-frequency VHC stimulation and in which LTP was induced by dMSA stimulation (“saporin” group). We analyzed the expression of genes 1 h after LTP induction (or 1.5 h after implantation of electrodes in the control group) in the ipsilateral dorsal and ventral hippocampi as well as the contralateral dorsal and ventral hippocampi.

First of all, it should be noted that the gene expression of the majority of the studied genes had higher expression in the ipsilateral (injured by the recording electrode) compared with the contralateral (intact) hippocampus in the control animals. As we noted in our previous study [12], this increase is likely to be a result of a spreading depression that occurred in the ipsilateral hippocampus after electrode implantation.

The first pair of analyzed genes included genes of neurotrophic factors *bdnf* (Figure 1A,C) and *ngf* (Figure 1B,D), since it was shown previously that their expression can be induced after MSA activation [16]. In the ipsilateral dorsal and ventral hippocampi, the induction of LTP resulted in a significant increase in the expression of *bdnf*. In the contralateral hippocampal parts, we did not observe any significant effect of either LTP or 192IgG-saporin treatment. The level of *ngf* mRNA appeared to be less sensitive to the conditions of our experiments. We found that only in the contralateral ventral hippocampus of the “saporin” group the level of *ngf* increased significantly compared with the control group, whereas LTP induction resulted only in the appearance of a trend toward an increase in the *ngf* level.

The second group of genes included immediate early genes that are frequently associated with LTP induction in the hippocampus (i.e., *arc, fos, egr1, egr2, egr3,* and *egr4*). Additionally, we studied the expression of *cyr61*, which was shown to be a gene that may potentially be modulated by the action of acetylcholine [17,18,19].

We found that *egr1* expression in the ventral contralateral hippocampus was significantly increased after LTP induction, and a cholinergic deficit prevented this increase (Figure 2A,B). In the dorsal contralateral hippocampus, we observed no changes in the expression of this gene after dMSA stimulation, and changes in the expression of *egr3* were similar (Figure 2C,D). LTP induction did not influence the expression of this gene in any studied hippocampal part. However, induction of a cholinergic deficit led to a significant decrease in the expression of this gene in the contralateral ventral hippocampus compared with both the LTP and control groups. In all other hippocampal parts, we observed only a trend toward a decrease in *egr3* mRNA in the saporin-treated group compared with other groups. 

Analysis of changes in the expression of *egr2*, *egr4*, and *fos* did not reveal any changes in the mRNA levels of these genes either after LTP induction or in the group with a cholinergic deficit (Appendix A).

All detected changes in the level of *arc* mRNA were at the level of trend according to Kruskall–Wallis ANOVA (*p* < 0.1), and this trend consisted of a decrease in the expression of this gene in the “saporin” group compared with the “LTP” group in the ipsilateral dorsal and ventral hippocampi as well as the contralateral dorsal hippocampus (Figure 2E,F).

Analysis of *cyr61* expression revealed significant changes in the dorsal hippocampus, where LTP induction led to an increase in the mRNA level of this gene, and this increase was absent in the animals with cholinergic deficits. Similar trends were observed in the ventral hippocampal parts (Figure 2G,H).

### 2.3. Inhibition of the M1 Muscarinic Receptors Alters the fEPSP Characteristics but Does Not Affect the Expression of Early Genes

Our next step was to analyze whether the changes in the expression of genes that we observed after the induction of LTP by dMSA stimulation were determined by the M1 subtype of muscarinic receptors. To this aim, we performed two series of experiments.

In the first experimental series, we studied the effects of the muscarinic antagonist pirenzepine on fEPSPs evoked by VHC stimulation in the CA1 area. The baseline was stable for at least 30 min before pirenzepine injection. Two-way ANOVA showed a significant effect of the drug on fEPSP amplitude (F1,13 = 5.8, *p* < 0.03). Significant differences were found for the time (F14,182 = 3.69, *p* < 0.0001) and time and drug interaction (F14,182 = 3.75, *p* < 0.0001), suggesting that a decrement in fEPSP amplitude over time is related to the pirenzepine action. Subsequent ANOVA for the separate conditions followed by post hoc testing revealed that pirenzepine induced a decrease in fEPSP amplitude, which became significant >50 min after the beginning of the injection (Figure 3A,C). The paired-pulse facilitation (PPF) during the baseline and after pirenzepine injection (30 msec interstimulus interval) showed significant intergroup differences (F1,13 = 4.72, *p* = 0.048), and the interaction between the group and time was insignificant (F14,182 = 1.71, *p* < 0.057). The post hoc test showed that the PPF ratio in the pirenzepine-treated rats was significantly higher compared with the control animals (Figure 3B,D). 

Furthermore, we performed analysis of the effect of a pirenzepine injection on the expression of genes. In order to do this, we compared three groups of animals: (1) rats implanted with electrodes in the VHC and CA1 area, which did not receive any stimulation, (2) rats in which the VHC was stimulated at a low frequency and saline was injected in the ventricle contralateral to the implanted electrode, and (3) rats in which the VHC was stimulated at a low frequency and pirenzepine was injected in the contralateral ventricle. Analysis of the gene expression 1 h after dMSA stimulation (~1.5 h after i.c.v. injection of pirenzepine (or saline)) showed that among the analyzed hippocampal parts and genes (*arc, bdnf, cyr61, egr1, egr3*, and *ngf*), low-frequency VHC stimulation caused a significant increase in the level of *bdnf* mRNA only in the ipsilateral ventral hippocampus, and this increase was not related to the activation of M1 receptors (Appendix A).

### 2.4. Inhibition of M1 Muscarinic Receptors Prevents LTP Induction and Induces Selective Suppression of the cyr61 Gene in the Dorsal Hippocampus

In the second series, we analyzed the effect of pirenzepine on LTP induction in the VHC-CA1 synapses after HFS of the dMSA and on the expression of genes that occurred after LTP induction. After a pirenzepine injection, the responses in the CA1 area to VHC stimuli were recorded for 40 min to confirm the stability of the baseline responses. Tetanic stimuli were then applied to the dMSA area to induce LTP. A comparison of the effect of HFS on the fEPSP amplitude in the control and pirenzepine-treated animals did not show a significant group effect (F1,17 = 2.24, *p* < 0.15). However, there was a significant time effect (F11,187 = 2.54, *p* = 0.005) and time and drug interaction (F11,187 = 2.149, *p* < 0.02). The post hoc test revealed that the rats from the control group had significantly higher amplitudes of fEPSPs compared with the baseline (Figure 4A,C). Two-way ANOVA did not show a significant effect of the drug on PPF (F1,19 = 1.6, *p* < 0.2), and no differences were observed for the time and drug interaction (F11,209 = 1.0, *p* < 0.44). A significant time effect was found for PPF (F11,209 = 2.25, *p* < 0.05). The PPF ratio significantly decreased over time independent of the experimental conditions. The PPF decrement in the pirenzepine-treated rats was insignificant (Figure 4B,D).

Analysis of the effects of pirenzepine on the gene expression after LTP induced by dMSA stimulation showed that the expression of the majority of the genes (*arc, bdnf, egr1, egr3, fos*, and *ngf*) was not altered in the ipsi- and contralateral dorsal hippocampi. In the ipsi- and contralateral ventral hippocampi, pirenzepine also did not alter most of the studied genes (*arc, bdnf, cyr61, egr1, fos*, and *ngf*) (Figure 5A–F; Appendix A). The only significant changes that were induced by pirenzepine included the suppression of *cyr61* expression in the contralateral dorsal hippocampus (Figure 5A) and an increase in *egr3* expression in the ipsilateral ventral hippocampus (Figure 5D).

## 3. Discussion

Our study was aimed at investigating which early genes are activated when LTP in the hippocampal synapses is induced by dMSA stimulation. We analyzed the effect of LTP induction in both the contra- and ipsilateral hippocampi to evaluate the role that may be played by a putative spreading depression occurring in the ipsilateral hippocampus after being damaged by the implantation of a recording electrode. We found that among the studied early genes, LTP induction led in the ipsilateral hemisphere to an increase in the expression of *bdnf* in both the dorsal and ventral hippocampal parts, an increase in *egr1* mRNA in the contralateral ventral hippocampus, and an increase in *cyr61* mRNA in the ipsilateral dorsal hippocampus. These increases in mRNA were absent when the LTP-inducing protocol was applied to animals with cholinergic deficit, pointing to the possibility that the observed changes in the mRNA expression may have resulted from the activation of medial septal cholinergic neurons. We further analyzed the possible role of M1 muscarinic receptors in changes in gene expression after LTP induction using an i.c.v. injection of pirenzepine. We found that LTP was not observed in the presence of pirenzepine, which is in agreement with previous studies where pirenzepine prevented LTP induction in the hippocampus [20,21,22]. However, the absence of LTP after the pirenzepine injection largely did not affect gene expression in the hippocampus. The only significant effect of pirenzepine was observed for the *cyr61* gene, where this drug suppressed *cyr61* expression in the dorsal contralateral hippocampus. The selective action of pirenzepine only in one hippocampal domain means that the increase in *cyr61* mRNA after LTP induction involves the activation of M1 receptors, but some other factors also play a role in this process. Presumably, the activation of M3 receptors, which are widely expressed in the hippocampus [23,24], may play some role.

Our results suggest that the processes underlying LTP development and the induction of early genes during LTP induction may be just parallel processes induced in the same cells by LTP-inducing synaptic activity. The induction of early genes after LTP induction in the hippocampus by different protocols was shown in many studies [25,26,27,28,29]. However, the role of the LTP-induced genes in the development of LTP was usually discussed in terms of induction of other processes important for LTP maintenance. Here, we show that LTP development may be prevented by pirenzepine but largely does not suppress gene expression after LTP induction. The latter means that LTP-inducing stimulation under conditions where LTP is blocked causes the same effect as under conditions where LTP is developed. Hence, LTP in hippocampal synapses per se is not a prerequisite for gene induction; rather, a certain type of synaptic signals (in our case, HFS of dMSA) is critical for enhancement of the expression of some early genes, and this enhancement can occur without LTP.

A comparison of the early gene expression in the ipsilateral and contralateral dorsal and ventral hippocampi showed a clear increase in the expression of immediate early genes in both ipsilateral hippocampal parts compared with the contralateral hippocampi. These results support our previous observations [12] that the implantation of electrodes leads to an increase in the majority of early response genes both at the site of implantation (dorsal ipsilateral hippocampus) and in the distant hippocampal part (ventral ipsilateral hippocampus) but not in the contralateral hippocampi. As we discussed previously [12], the most likely explanation for this nonlocal effect in the ipsilateral hippocampus is a wave of spreading depression (currently, some researchers also call it spreading depolarization) which is initiated at the site of the hippocampal damage by the implanted electrode and spreads over the entire ipsilateral hippocampus. The occurrence of a spreading depression in one of the hippocampi helps us to detect additional changes that are otherwise hidden. It was shown previously that a spreading depression can change the DNA methylation pattern [30] and the modification of histones [31]. These epigenetic shifts may underlie some changes that we observed in our study. Presumably, a depolarization wave, which spread over the entire ipsilateral hippocampus, not only induced the expression of early genes but also changed the epigenetic state of the hippocampal cells and their readiness to respond to dMSA stimulation. This may be seen in the example of *bdnf*, whose expression increased only in the ipsilateral hippocampus after high-frequency dMSA stimulation. This result suggests that *bdnf* expression is controlled by some additional factors that have to be activated and/or inactivated before it can be changed by synaptic activity. Importantly, the induction of a cholinergic deficit did not prevent the increase in the *bdnf* level after dMSA stimulation, suggesting that dMSA stimulation may induce the expression of some genes independent of cholinergic projections from the medial septum.

Another example of a gene whose response to dMSA HFS was changed in the ipsilateral hippocampi compared with the contralateral parts is *egr1*. LTP induction led to an increase in *egr1* expression only in the contralateral hippocampus. Presumably, an increase in *egr1* mRNA caused by a spreading depression after electrode implantation in the ipsilateral hippocampus prevented a further increase in the level of this gene and altered the readiness of this gene for transcription after dMSA stimulation. Pirenzepine application did not cause any decrease in any hippocampal part, suggesting that an increase in the mRNA level of this early gene is not related to the M1 receptors.

LTP induction did not lead to an increase in *egr3* expression in any hippocampal parts. However, LTP induction in animals with cholinergic deficits caused a significant decrease in the expression of this gene in the contralateral ventral hippocampus and a similar trend in other hippocampal parts. At the same time, LTP induction in the presence of pirenzepine led to an increase in the expression of this gene in the ipsilateral ventral hippocampus. In our opinion, these data point to a highly complex regulation of this gene in the hippocampal cells. Our data and previous transcriptomic studies [4,7] show that *egr3* expression under resting conditions is not influenced by cholinergic input from the septum and does not depend on the basal activity of the M1 muscarinic receptors. However, dMSA stimulation under the conditions of cholinergic degeneration suppresses *egr3* expression. It is worth reminding here that the medial septal area contains cholinergic, GABAergic, and glutamatergic neurons, all of which project to the hippocampus [32,33]. Activation of this area by electrical stimulation leads to the activation of all neuronal subpopulations and, in the case of cholinergic degeneration, leads to the activation of GABAergic and glutamatergic neurons, which may finally result in the inhibition of hippocampal neurons because septal glutamatergic neurons predominantly innervate hippocampal inhibitory interneurons [32]. Presumably, the strong inhibitory drive that occurs during HFS of dMSA under the conditions of a cholinergic deficit may be one of factors that suppresses the expression of several genes which did not respond to HFS of dMSA under normal conditions (*egr3* and *arc*). However, in the case of *egr3*, we also found that its transcription was induced after inhibition of the M1 muscarinic receptors in the ventral ipsilateral hippocampus (with a trend in the dorsal ipsilateral part) after application of HFS to dMSA. Note that M1 receptors may be coupled not only to the activating Gq protein, as stated in many studies, but also to the inhibitory Gi/o protein [34]. Presumably, in our experimental conditions, when a spreading depression in the ipsilateral hippocampus shifted the epigenetic landscape, the application of pirenzepine suppressed not only activation via the Gq protein but also its inhibition via the Gi/o protein and led to an increase in *egr3* expression.

Our study has several important limitations. First, we used urethane to achieve long-term anesthesia for the implantation of electrodes and electrophysiological recordings. Urethane anesthesia is known to activate the sympathetic system, which leads to the elevation of plasma catecholamines, glucose, insulin [35], and CO_2_ saturation of the blood and a decrease in body temperature [36]. Presumably, these changes may serve as additional epigenetic factors that can alter the readiness of hippocampal cells to the induction of transcriptional changes after certain stimuli. Our current and previous [12] data suggest that changes in the expression of early genes caused by putative spreading depressions after electrode implantation in the hippocampus were similar to the changes observed in non-anesthetized animals [37]. This comparison suggests that urethane is likely to have a negligible effect on the detected changes in the expression of early genes, but we cannot completely exclude this influence. Currently, it is not clear whether the mentioned urethane-induced changes in body metabolism may affect the picture of gene expression in the brain tissue and, in particular, in the hippocampus. Further studies are necessary to clarify this issue.

Another limitation of our study is the interpretation of the results obtained in the experiments with 192IgG-saporin. The problem with these results is that their interpretation requires, as we mentioned above, a detailed description of the septal projections activated during the stimulation of dMSA, which currently are not available. Our data suggest that septal cholinergic projections, which are present in dMSA, definitely play a role in the induction of several early genes after LTP induction by dMSA stimulation, whereas the expression of other early genes does not depend on cholinergic projections. The role of other (probably GABAergic and glutamatergic) projections remains obscure and requires more detailed studies.

Previous data suggest that LTP induction [25,27,38,39,40,41] and the activation of M1 receptors [19,38] may induce the expression of various early genes. Here, we showed that LTP induced by HFS of dMSA resulted in an increase in the expression of *egr1* and *cyr61* but not *arc*, *bdnf*, or *fos*. This increase was dependent on the presence of cholinergic septal neurons but independent of the activation of the M1 receptors. Taken together with previous data, it is possible to conclude that the protocol of LTP induction, the involvement of neurons with different ergicities, and different postsynaptic receptors may strongly influence the outcome at the level of gene transcription. According to our data, cholinergic dMSA input into the hippocampus is critical for both LTP and transcriptional activation of some early genes. However, septal glutamatergic and GABAergic neurons projecting to the hippocampus [32,33] seem to play an important role in the induction of gene expression in the hippocampal cells but not in LTP induction by dMSA HFS. It remains unclear from our data whether the changes in gene expression after dMSA HFS were a result of the activation of one of the mentioned components of dMSA input into the hippocampus or resulted from the interplay between the glutamatergic, GABAergic, and cholinergic components. More delicate studies on the dissection of the cellular and receptor mechanisms of induction of gene expression after dMSA HFS are required.

In terms of pathology, our data suggest that a cholinergic deficit per se is not a substantial factor for its development. However, when a complex signaling network, which in a healthy brain includes the cholinergic neurons of the basal forebrain, becomes activated under the conditions of a cholinergic deficit, it may lead to a seriously distorted response, because we showed here that a similar pattern of activation of the same groups of neurons may give the opposite result at the level of gene expression and, as a consequence, unexpected long-term changes in function at the network level.

Our study has several important outcomes. First, the LTP-inducing activity of dMSA leads to the induction of a number of genes in the hippocampus. Second, the induction of some of these genes depends on the activation of septal cholinergic neurons. Third, blockage of the M1 receptors prevents LTP induction by dMSA stimulation but does not affect the induction of early genes, pointing to an important role played by M1 receptors in LTP induction but not the induction of early genes. Taken together, our data suggest that LTP induction by dMSA stimulation is accompanied by the enhancement of early gene expression but is not a prerequisite for this enhancement.

## 4. Materials and Methods

Adult male Wistar albino rats (250–300 g) were received from the Research Center of Biomedical Technology’s nursery “Pushchino”. A total of 81 rats were involved in the study. The animals were housed under standard conditions at 21 ± 1 °C with a 12 h light/dark cycle. Food and water were provided ad libitum. All experiments were performed in accordance with the ethical principles stated in the European Directive (2010/63/EU) and were approved by the ethical committee of the Institute of Higher Nervous Activity and Neurophysiology of the Russian Academy of Sciences.

### 4.1. 192IgG-Saporin Injection

To induce a cholinergic deficit, 192IgG-saporin was injected into the medial septum area (MSA) with a microinfusion pump (Stoelting Co., Wood Dale, IL, USA) at a rate of 0.2 μL/min using a Hamilton syringe (Hamilton Company, Reno, NV, USA) as previously described in [11]. The rats were anesthetized using chloral hydrate (400 mg/kg, i.p.). For additional anesthetic and analgesic effects, the ears and wound surface of each animal were treated with lidocainum (Pharmstandart, Moscow, Russia). The stereotaxic coordinates were as follows: +0.4 mm anteroposterior, −1.5 mm lateral, and 14° [42]. The drug was infused at a concentration of 1.5 μg/per rat. The rats were allowed to recover for 21 days. The injection sites were confirmed for each experiment. The animals, in which the number of cholinergic neurons in the medial septal area after 192IgG-saporin injection was >50% that of the controls, were excluded from further experiments.

### 4.2. RNAseq

The total RNA was isolated from the hippocampal samples using ExtractRNA Reagent (Evrogen, Moscow, Russia) following the manufacturer’s protocol. The extracted RNA in the samples was analyzed using an Agilent 2100 Bioanalyzer to confirm the purity of the RNA isolation. In all samples, RIN > 8. For RNA-seq analysis, we took three rats from the control and 192IgG-saporin-treated groups of animals as previously described in [7].

For the depletion of ribosomal RNA, we used an NEBNext^®^ rRNA Depletion Kit in accordance with the manufacturer’s protocol. Then, we prepared cDNA libraries using an Ion Total RNA-Seq Kit v2 for Whole Transcriptome Libraries (Ion Torrent, Life Technologies, Burlington, ON, Canada), following the manufacturer’s protocols as previously described in [7].

Sequencing was performed using an Ion PITM Sequencing 200 Kit v3 and an Ion PITM Chip v2 on an Ion ProtonTM sequencer. One chip contained four libraries with different barcodes.

The quality of the raw single-end reads was assessed using FastQC 0.12.0 (https://www.bioinformatics.babraham.ac.uk/projects/fastqc/ (accessed on 1 March 2023)). High–quality reads were filtered by eliminating low-quality reads with Q < 20 from the raw reads. To evaluate the quality of trimming and removal of adapters, the fast and high-performance tool “bbduk” was used (https://jgi.doe.gov/data-and-tools/software-tools/bbtools/bb-tools-user-guide/bbduk-guide/ (accessed on 1 March 2023)). Furthermore, the RNA-seq read mapper STAR [43] was used to align the reads against the 7.2 version of the rat genome [44]. In total, 54.9–65.4% of the reads were mapped to genomes. The tool feature Counts from the Subread package (v.1.6.5) was used for counting the reads mapped to genomic features [45]. Reads that mapped to multiple locations were removed from the analysis. Approximately 44.8–50.4% of the reads mapped uniquely to different genomic features.

Differentially expressed genes were called using the DESeq2 package (version 4.3) [46], an R package implementing a model based on a negative binomial distribution which was developed in order to cope with biological variance. The software was run under R release 4.3.0. In this type of analysis, DE genes were selected if the adjusted *p* value of the DeSeq2 test was <0.1. To estimate the dispersion trend and apply a variance-stabilizing transformation, the vst function from DESeq2 was used. A gene expression principal component analysis (PCA) plot, which provides a map of the distances between samples, was generated using the plotPCA function from DESeq2 package in R.

### 4.3. RNA Isolation and Reverse Transcription

In all experiments, tissue samples were collected 1 h after the start of stimulation, placed in 1.5 mL tubes, and frozen in liquid nitrogen. RNA isolation was performed using an ExtractRNA reagent (Evrogen, Moscow, Russia) in accordance with the manufacturer’s recommendations. To remove traces of genomic DNA, the RNA samples were treated with DNase I (Thermo Scientific, Vilnius, Lithuania). Reverse transcription was performed using the MMLV RT reagent kit (Evrogen, Moscow, Russia) and murine RNase Inhibitor (New England Biolabs, Ipswich, MA, USA) as recommended by the manufacturers. An equimolar mixture of random decaprimer (Evrogen, Moscow, Russia) and oligo(dT)15 primer (Evrogen, Moscow, Russia) was used, and the concentration of each primer in the reaction was 1 μM. After reverse transcription, the reaction mixture was diluted eightfold with deionized water.

### 4.4. qPCR

The relative quantities of the mRNA for the genes of interest were evaluated with a BioRad CFX384 real-time PCR station (BioRad, Singapore) using a qPCRmixHS SYBR + LowROX mix for PCR (Evrogen, Moscow, Russia) according to the manufacturer’s recommendations. The relative quantities of mRNA were normalized to the geometric mean of the mRNA expression levels for the ywhaz and osbp genes. The quality of the DNase treatment was evaluated in all the samples and genes by performing a negative control qPCR with the product of DNase I treatment. The primers for qPCR were taken from the previous study [12]. Gene expression was analyzed by the E−ΔΔCt method. In the experimental series with 192IgG-saporin and analysis of the effect of pirenzepine under basal conditions, the qPCR results were normalized to the external reference sample. In the experimental series with analysis of the effect of pirenzepine on changes in gene expression after LTP induction, the data were normalized to a sample of the ipsilateral ventral hippocampus in the group treated with pirenzepine. After analysis of gene expression, two animals were excluded from analysis in the first series of experiments due to abnormal (extremely low or high) expression of the majority of genes in the samples of a given animal.

### 4.5. Electrophysiology Recording and Drug Injection

During the recording session, the rats were anesthetized with urethane (1.75 g/kg, intraperitoneally) and fixed in the stereotaxic frame. The fEPSP was found using previously described methods [11]. In brief, a stimulated nickel-chrome electrode (diameter: 80 µm) was lowered into the ventral hippocampal commissure (VHC) (−1.3 mm anteroposterior, −1.0 mm lateral, approximately 3.5 mm ventral to the dura) and in the MSA (+0.4 mm anteroposterior, −1.5 mm lateral, 14°). A recording electrode was implanted into the CA1 area (−2.7 mm anteroposterior; −1.5 mm lateral, approximately 2.2 mm ventral to the dura) [47]. One electrode under the skin served as a ground and as a reference electrode. All electrodes were fixed to the skull using quick-setting dental plastic (protakril M).

The fEPSP amplitude in the CA1 field evoked by stimulation of the VHC (30 msec interstimulus interval, 20 s intertrain time at an intensity of 100–400 μA) was obtained from 10 successive stimuli and recorded every 10 min. The test’s paired pulse intensity was adjusted to evoke 40–50% of the maximum amplitude response. For the basal synaptic transmission experiments, the recordings were continued for 30 min before pirenzepine administration and for 1 h after the drug’s injection to examine the effects of pirenzepine on basal synaptic transmission. For the LTP experiments, LTP was induced 40 min after pirenzepine (or saline) administration by high-frequency stimulation of the dMSA (HFS, 5 series, 4 trains of 5 stimuli with a frequency of 100 Hz, an intertrain interval of 200 ms, and an inter-series interval of 30 s). Post-LTP recordings were performed for 1 h. Urethane anesthesia is used for non-recovery procedures of an exceptionally long duration, where preservation of autonomic reflexes is essential and thus does not need any additional euthanasia procedure.

In the experiments with intracerebroventricular (i.c.v.) injections of pirenzepine (50 nmol/rat, 1 uL), a Hamilton syringe was lowered into the contralateral ventricle (−0.8 mm anteroposterior, −1.5 mm lateral, approximately 3.8 mm ventral to the dura) [47].

### 4.6. Immunohistochemistry

To control the death of septal cholinergic neurons after the injection of 192IgG-saporin, at the end of experiments, the frontal parts of the brain were fixed using 4% paraformaldehyde solution and then sliced to confirm cholinergic lesions in the MSA [7]. For the ChAT immunoreaction, the 50 μm-thick coronal brain sections of the medial septum area were incubated in a rat anti-ChAT primary antibody (rabbit anti-Choactase 1:500, Santa Cruz Biotechnology, Santa Cruz, CA, USA) at 4 °C overnight. Subsequently, the sections were incubated with secondary antibodies (1:800, goat anti-rabbit IgG-biotin, Sigma–Aldrich, St. Louis, MO, USA) in blocking solution at room temperature for 2 h. After additional washing in PBS, the sections were incubated with avidin–biotin–HRP complex (ABC Elite kit, Vector Labs, Mowry Ave Newark, CA, USA) for 1 h, and 3,3′-diaminobenzidine (SIGMA-Fast Kit, Sigma–Aldrich, St. Louis, MO, USA) was used as a chromogen for the development of staining. All images were acquired with a Leica DM6000B microscope (Leica, Wetzlar, Germany) or Keyence BZ-9000 (Keyence, Osaka, Japan). Delineation of the brain structures was performed in the images according to Paxinos and Watson’s atlas (1998). All stained cells within the medial septum and diagonal band of Broca were counted. Only cells located within one focal plane were counted to prevent overestimation. The mean number of cells per section was counted and considered a representative value for one animal. The number of cells in an arbitrarily selected control animal was taken to be 100%. Data on 3 out of 10 animals in the “saporin” group were excluded from further analysis due to the absence of loss of ChAT-positive neurons in the MSA after 192IgG-saporin’s injection. A comparison of the number of cells between the “control” (*n* = 7), “LTP” (*n* = 8), and “saporin” (*n* = 7) groups revealed a significant (F(2,19) = 30,536; *p* = 0.000001; ANOVA) decrease in the number of ChAT-positive cells in the MSA after 192IgG-saporin injection (100 ± 6.4%, 105 ± 10%, and 22 ± 7%, respectively).

### 4.7. Statistical Analysis

All electrophysiological data are presented as the mean ± SEM. Across the groups of electrophysiological data, statistical significance between the means was determined using a mixed-design analysis of variance followed (where applicable) by Fisher’s least significant difference (LSD) post hoc test to reveal group differences in separate time intervals.

All qPCR data did not follow a normal distribution and were analyzed in R using a *wilcox_test* for comparison of two groups and *kruskal.test* for comparison of three groups. Differences were considered significant at *p* < 0.05, while those at 0.05 ≤ *p* < 0.1 were considered trends. A Dunn test (*dunn.test*) with Benjamini–Hochberg correction for multiple comparisons was used for post hoc comparisons. Differences were considered significant at *p_adjusted_* < 0.05 and trends at 0.05 ≤ *p_adjusted_* < 0.1. Plots were constructed using ggplot2 in R [48].

## Figures and Tables

**Figure 1 ijms-24-12970-f001:**
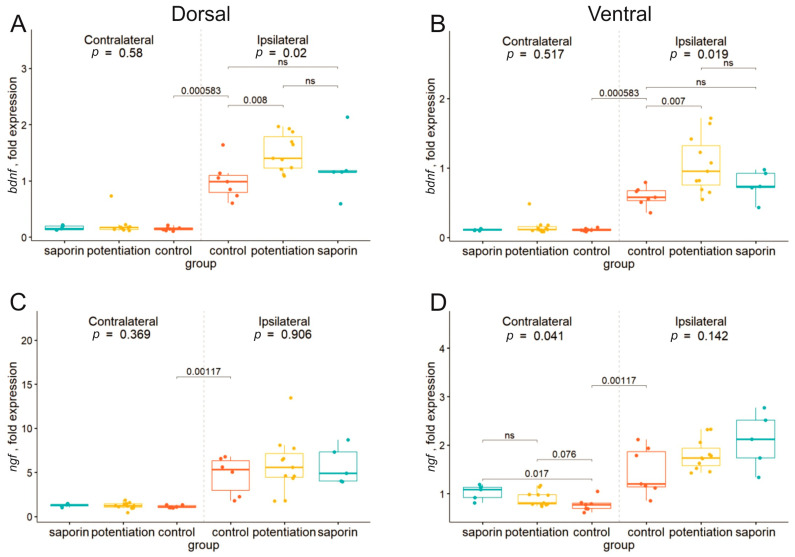
Changes in the expression of *bdnf* (**A**,**C**) and *ngf* (**B**,**D**) genes in the dorsal (**A**,**B**) and ventral (**C**,**D**) parts of the left (contralateral) and right (damaged by the implanted electrode) hippocampi after high-frequency stimulation of the dorsal medium septal area in the control animals (potentiation) and animals with degeneration of cholinergic neurons in the medial septal area (saporin). The *p* values are shown for a Kruskal–Wallis test for comparison of the three groups in one hemisphere. Interhemispheric comparisons were performed only in the control group using the Mann–Whitney test. For the intergroup comparisons, *p_adjusted_* was calculated using the Benjamini–Hochberg method. Differences were considered significant at *p_adjusted_* < 0.05. For interhemispheric comparisons, exact *p* values are shown.

**Figure 2 ijms-24-12970-f002:**
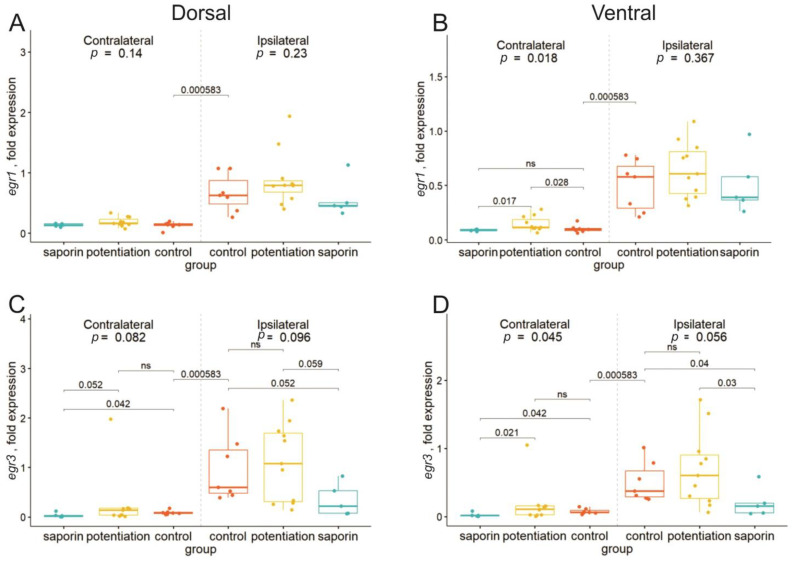
Changes in the expression of early genes in the dorsal (**A**,**C**,**E**,**G**) and ventral (**B**,**D**,**F**,**H**) parts of the left (contralateral) and right (damaged by the implanted electrode) hippocampi after high-frequency stimulation of the dorsal medium septal area in the control animals (potentiation) and animals with degeneration of cholinergic neurons in the medial septal area (saporin). The *p* values are shown for a Kruskal–Wallis test for comparison of the three groups in one hemisphere. Interhemispheric comparisons were performed only in the control group using the Mann–Whitney test. For the intergroup comparisons, *p_adjusted_* was calculated using the Benjamini–Hochberg method. Differences were considered significant at *p_adjusted_* < 0.05. For interhemispheric comparisons, exact *p* values are shown.

**Figure 3 ijms-24-12970-f003:**
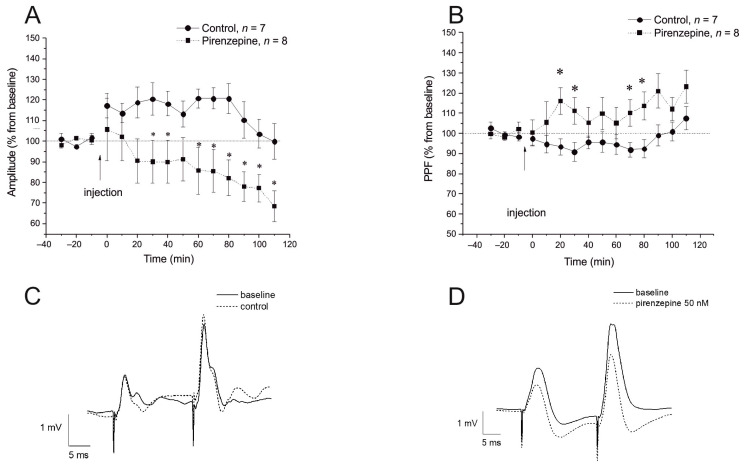
Effect of pirenzepine on field excitatory postsynaptic potential (fEPSP) in the hippocampal CA1 region. (**A**) The time course of fEPSP in pirenzepine (*n* = 8) and control groups (*n* = 7) for 120 min after intracerebroventricular (i.c.v.) injection of pirenzepine. (**B**) The time course of PPF for 120 min after i.c.v. injection of pirenzepine. (**C**,**D**) The fEPSP in vivo varies as a function of the drug treatment in the intact rat CA1 region. Each point represents the mean ± S.E.M. percentage of basal fEPSP amplitude or PPF at 0 min. * Significant differences against the control group, where *p* < 0.05.

**Figure 4 ijms-24-12970-f004:**
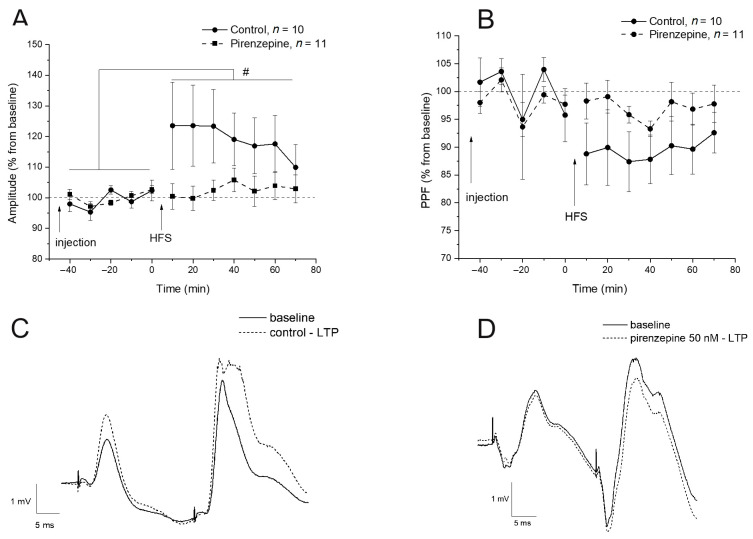
Effect of pirenzepine on long-term potentiation (LTP) in the synapses formed by the ventral hippocampal commissure on the hippocampal CA1 neurons. (**A**) The time course of fEPSP in pirenzepine (*n* = 11) and control groups (*n* = 10) for 70 min after an intracerebroventricular (i.c.v.) injection of pirenzepine and high-frequency stimulation (HFS). (**B**) The time course of PPF for 70 min after i.c.v. injection of pirenzepine and HFS. (**C**,**D**) The fEPSP in vivo varied as a function of the drug treatment in the intact rat CA1 region. Each point represents the mean ± S.E.M. percentage of basal fEPSP amplitude or PPF at 0 min. # *p* < 0.05, with significant differences compared with baseline.

**Figure 5 ijms-24-12970-f005:**
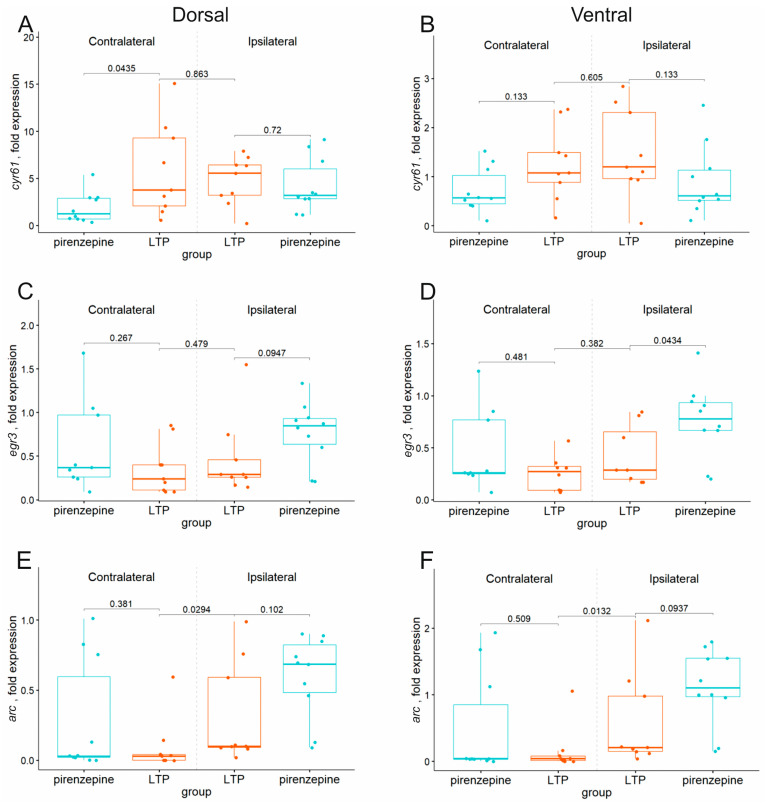
Changes in the expression of early genes in the dorsal (**A**,**C**,**E**) and ventral (**B**,**D**,**F**) parts of the left (contralateral) and right (damaged by implanted electrode) hippocampi after high-frequency stimulation of the dorsal medium septal area in the animals (LTP) intracerebroventricularly treated with saline (LTP) or pirenzepine (pirenzepine), with *p* values shown for Mann–Whitney test for comparison in one hemisphere.

## Data Availability

Raw data are available from the authors upon request.

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
