# Peer review of "The Induction of Long-Term Potentiation by Medial Septum Activation under Urethane Anesthesia Can Alter Gene Expression in the Hippocampus"

_ijms, 2023, doi:10.3390/ijms241612970_

Round 1

Reviewer 1 Report

This study from Dobryakova et al, is quite straightforward and aims to define the changes in gene expression induced by stimulation of cholinergic neurons in the MSA, and investigate the relationship between the cholinergic activity in the dorsal MSA and hippocampal neurons, and if this relationship is dependent on cholinergic MSA neurons. Unfortunately, the report shows a few major defects.

The authors mention a bulk RNA sequencing in results and methods but do not show any results related to that experiment. The authors show that dMSA cholinergic neurons stimulation is associated with gene expression changes in the hippocampus, however it is not clear how the authors chose neurotropic target genes. Furthermore the study looks underpowered and the saponin group results are inconclusive. All qPCR data are reported as fold change, but it is not clear which one is the reference group used for each experiment.

Finally, the conclusion does not really give an explanation of the data presented and is limited to a generic involvement of unclear ‘various signals’.

The manuscript itself needs some revision (i.e. line 206-208 seems to be the comment of an author), sometimes the aims are not very clear and it is difficult to follow the flow of the results.

Author Response

We are grateful to reviewers for their comments and suggestions. They helped us to improve our manuscript and find other errors that were not addressed by the referees but are still important for correct data presentation. In addition to amendments that were proposed by the reviewers, we also found mistakes made by us during analysis of the data during preparation of the manuscript. In the first series of manuscript (experiments with injections of 192IgG-saporin), one sample in the dorsal contralateral hippocampus in the “control” group and one sample in the dorsal contralateral hippocampus in the “potentiation” group were lost during data processing, which led to miscalculation of some statistical data in the dorsal contralateral hippocampus in this series of experiments. In addition to this mistake, the significance of differences between the ipsilateral and contralateral hemispheres were also recalculated because p-values shown in the previous version of the manuscript were also calculated incorrectly due to technical errors. We corrected these errors and apologize for them. In general, they did not significantly affect conclusions of the manuscript and we made practically no changes in the text due to this error; however, we believe that correct presentation of data is important for scientific community. Again, we grateful for reviewers for their help in improving manuscript.

Below, our responses to reviewer's suggestions are given in Italics. 

The authors mention a bulk RNA sequencing in results and methods but do not show any results related to that experiment.

The data on differential expression were included as Supplementary Table. Additionally, we constructed PCA plot to show the absence of differences between the samples of different groups.

 The authors show that dMSA cholinergic neurons stimulation is associated with gene expression changes in the hippocampus, however it is not clear how the authors chose neurotropic target genes.

The choice of neurotrophin-expressing genes is based on previous publication by Lindefors et al., 1992 (Septal Cholinergic Afferents Regulate Expression of Brain-Derived Neurotrophic Factor and Beta-Nerve Growth Factor mRNA in Rat Hippocampus. Exp. Brain Res.) where injection of potent glutamate agonist quisqualate to the medial septal area leads to postponed elevation of mRNA of bdnf and ngf in the hippocampus.

Furthermore the study looks underpowered and the saponin group results are inconclusive.

This suggestion of reviewer is unclear. However, we added description of limitations of our study to the discussion part.

 All qPCR data are reported as fold change, but it is not clear which one is the reference group used for each experiment.

In the experimental series with 192IgG-saporin and analysis of effect of pirenzepine under basal conditions, qPCR results were normalized to the external reference sample. In the experimental series with analysis of effect of pirenzepine on changes in gene expression after LTP induction, the data were normalized to a sample of ipsilateral ventral hippo-campus in the group treated with pirenzepine. This description of reference samples was added to Materials and methods.

Finally, the conclusion does not really give an explanation of the data presented and is limited to a generic involvement of unclear ‘various signals’.

We added to the discussion the final part where we summarized our findings and made compact conclusions.

The manuscript itself needs some revision (i.e. line 206-208 seems to be the comment of an author), sometimes the aims are not very clear and it is difficult to follow the flow of the results.

Thank you. This was corrected

Reviewer 2 Report

Using a rat model, this manuscript explored the effects of stimulation of the dMSA on the expression of genes and electrophysiology. I found some parts need to be improved and some mistakes need to be fixed.

1. The title of this manuscript needs to be modified to cover the whole study.

2. In Abstract part, I don't think the conclusion is Line 19-21 is sufficiently accurate.

3. What do you mean by saying the higher expresssion in the ipsilateral is likely to be a result of spreading depression in Line 91-92?

4. In Fig.1, there is an extra arrow at the bottom of the figure. Besides, please label individual images in each figure to A,B,C..., then you can describe the result much clearly, just like Fig.3.

5. The legend for Fig.3 is completely wrong.

6. Sentences in line 206-208 are redundant.

7. Show us the RNA-seq result, even through there's no significant changes.

8. I didn't find any immunohistochemistry related result in current manuscript, why do you have "Immunohistochemistry" in Method part?

9. In Line 418-421, what do "()" mean?

10. I think you can adjust the titles of the Result part to the specific result or conclusion, which can make it much easier to understand.

11. I found some grammar problem, such as Line 36, 294.

12. I also found missing  punctuation, such as line 51, 52, 127.

13. Give us the full name, when the abbreviation appears at the first time, such as Line 81.

  •  

Need to be improved.

Author Response

We are grateful to reviewers for their comments and suggestions. They helped us to improve our manuscript and find other errors that were not addressed by the referees but are still important for correct data presentation. In addition to amendments that were proposed by the reviewers, we also found mistakes made by us during analysis of the data during preparation of the manuscript. In the first series of manuscript (experiments with injections of 192IgG-saporin), one sample in the dorsal contralateral hippocampus in the “control” group and one sample in the dorsal contralateral hippocampus in the “potentiation” group were lost during data processing, which led to miscalculation of some statistical data in the dorsal contralateral hippocampus in this series of experiments. In addition to this mistake, the significance of differences between the ipsilateral and contralateral hemispheres were also recalculated because p-values shown in the previous version of the manuscript were also calculated incorrectly due to technical errors. We corrected these errors and apologize for them. In general, they did not significantly affect conclusions of the manuscript and we made practically no changes in the text due to this error; however, we believe that correct presentation of data is important for scientific community. Again, we grateful for reviewers for their help in improving manuscript.

Below, our responses to reviewer's suggestions are given in Italics. 

  1. The title of this manuscript needs to be modified to cover the whole study.

The title was corrected.

  1. In Abstract part, I don't think the conclusion is Line 19-21 is sufficiently accurate.

We changed the conclusion.

  1. What do you mean by saying the higher expresssion in the ipsilateral is likely to be a result of spreading depression in Line 91-92?

It is known that spreading depression induces a strong shift in the expression of immediate early genes. Acute electrode implantation in our case causes tissue damage which is known to induce wave of spreading depression. Spreading depression is known to induce expression of many early genes (please, see discussion of this issue in our previous report Spivak et al., 2022 IJMS). Since here we did not record directly spreading depression, we just refer here to previous study and give the most probable explanation of the observed difference between the injured and uninjured hippocampi.

  1. In Fig.1, there is an extra arrow at the bottom of the figure. Besides, please label individual images in each figure to A,B,C..., then you can describe the result much clearly, just like Fig.3.

Thank you. The figures and legends were corrected as suggested by the reviewer.

  1. The legend for Fig.3 is completely wrong.

Thank you. The legend was corrected

  1. Sentences in line 206-208 are redundant.

Thank you. The lines were removed.

  1. Show us the RNA-seq result, even through there's no significant changes.

The data on differential expression were included as Supplementary Table. Additionally, we constructed PCA plot to show the absence of differences between the samples of different groups.

  1. I didn't find any immunohistochemistry related result in current manuscript, why do you have "Immunohistochemistry" in Methodpart?

We performed immunochemistry staining in experiments where 192IgG-saporin was used to induce death of cholinergic septal cells. This is necessary to confirm death of cholinergic cells after injection of the immunotoxin. In our case, we removed 3 out 10 animals where loss of cholinergic neurons after 192IgG-saporin was less than 50%. The data on death of cholinergic cells in the medial septum after 192IgG-saporin were inserted in the Method part, since they are of technical character. The statement on exclusion of animals were also included in the Method part.

  1. In Line 418-421, what do "()" mean?

This is just representation of commands that were used in R for calculation of statistical results. We decided to remove ().

  1. I think you can adjust the titles of the Result part to the specific result or conclusion, which can make it much easier to understand.

We corrected the titles in the Result part in accordance with your comment.

  1. I found some grammar problem, such as Line 36, 294.

Thank you. We corrected the sentences.

  1. I also found missing  punctuation, such as line 51, 52, 127.

Thank you. The punctuation was corrected.

  1. Give us the full name, when the abbreviation appears at the first time, such as Line 81.

Thank you. This was corrected.

Reviewer 3 Report

This authors experimented the changes in the expression of early genes in hippocampal cells leading to long-term potentiation in the hippocampus. This study seems good.

However, you must explain the study design in detail.

First of all, it uses urethane, but it is better not to use urethane according to the current animal ethics code. you have to explain why you use urethane as anesthesia.

All experiments were performed in accordance with 305 the ethical principles stated in the European directive (2010/63/EU) and were approved by the Ethical Committee of the Institute of Higher Nervous Activity and Neurophysiology of the Russian Academy of Sciences. It is listed in the MM, but the approval number must be included. Also, please explain whether the use of urethane is approved at this time.

The use of urethane induces acetylcholine, adrenaline, etc. throughout the body by acting on the sympathetic nervous system. Please explain whether these effects can be eliminated from the results of this study.

Some experimental data have been investigated contralaterally and ipsilaterally, but the significance of the experimental data and how to evaluate its conclusions neuroanatomically should be explained in detail. .

Author Response

We are grateful to reviewers for their comments and suggestions. They helped us to improve our manuscript and find other errors that were not addressed by the referees but are still important for correct data presentation. In addition to amendments that were proposed by the reviewers, we also found mistakes made by us during analysis of the data during preparation of the manuscript. In the first series of manuscript (experiments with injections of 192IgG-saporin), one sample in the dorsal contralateral hippocampus in the “control” group and one sample in the dorsal contralateral hippocampus in the “potentiation” group were lost during data processing, which led to miscalculation of some statistical data in the dorsal contralateral hippocampus in this series of experiments. In addition to this mistake, the significance of differences between the ipsilateral and contralateral hemispheres were also recalculated because p-values shown in the previous version of the manuscript were also calculated incorrectly due to technical errors. We corrected these errors and apologize for them. In general, they did not significantly affect conclusions of the manuscript and we made practically no changes in the text due to this error; however, we believe that correct presentation of data is important for scientific community. Again, we grateful for reviewers for their help in improving manuscript.

Below, our responses to reviewer's suggestions are given in Italics. 

First of all, it uses urethane, but it is better not to use urethane according to the current animal ethics code. you have to explain why you use urethane as anesthesia.

Urethane provides stable long-term terminal anesthesia, which is frequently used by researchers in different laboratories (for example, see recent studies Rusheen et al., 2023 Brain; Lao-Rodriguez et al., 2023 Sci. Adv.; Johnson et al., 2023 Hippocampus). In our case, we performed long-term experiments which required anesthesia with duration of more than 2 h (especially in the case of experiments with pirenzepine). In addition, this long-term anesthesia gave us opportunity to collect hippocampal samples at the end of long-term experiments without applying additional anesthesia.

All experiments were performed in accordance with 305 the ethical principles stated in the European directive (2010/63/EU) and were approved by the Ethical Committee of the Institute of Higher Nervous Activity and Neurophysiology of the Russian Academy of Sciences. It is listed in the MM, but the approval number must be included. Also, please explain whether the use of urethane is approved at this time.

Ethical Committee of our Institute asks to provide all details of experiments including drugs that are used for anesthesia. In case of approval, protocol issued by the Committee approves all manipulations with the animals; in our case, urethane anesthesia was approved.

The use of urethane induces acetylcholine, adrenaline, etc. throughout the body by acting on the sympathetic nervous system. Please explain whether these effects can be eliminated from the results of this study.

The issue of additional effects of urethane is very important. Therefore, we added additional discussion of this issue to the text of manuscript in Discussion part as a limitation of our study.

Some experimental data have been investigated contralaterally and ipsilaterally, but the significance of the experimental data and how to evaluate its conclusions neuroanatomically should be explained in detail. .

We added description to the Discussion part.

Round 2

Reviewer 1 Report

The manuscript's progress is evident, yet certain pivotal aspects warrant attention. To amplify its comprehensiveness, a more expansive scrutiny of RNA sequencing data is warranted—encompassing not just isolated gene selection but also broader biological processes and gene networks pertinent to distinct groups, especially concerning LTP establishment. Additionally, the discourse on mechanisms interlinking LTP with gene expression merits further elaboration, particularly when contrasting cholinergic, GABAergic, and glutamatergic neurons. It is worth noting that in line 318, the mention of the 'epigenetic landscape' lacks clarity regarding its specific reference. If this pertains to a potential explanation for gene expression changes, a more detailed exploration is crucial. Furthermore, citing research that links depolarization to alterations in the epigenetic profiles of various neuronal types would strengthen the manuscript's assertions.

Author Response

We are grateful to reviewer for comments and suggestions that helped us to improve our manuscript. Below, our replies to reviewer’s suggestions are given in bold.

To amplify its comprehensiveness, a more expansive scrutiny of RNA sequencing data is warranted—encompassing not just isolated gene selection but also broader biological processes and gene networks pertinent to distinct groups, especially concerning LTP establishment.

We agree with reviewer that RNA-sequencing data are usually analyzed in terms of biological processes and gene networks. However, it is worth to note that this sort of analysis is performed when some groups of genes are isolated from the entire set of RNA-seq data (usually, on the basis of differential expression). In our case, we have a situation when a comparison of two experimental groups (control and 192IgG-saporin-treated) did not reveal significant differences between them. In this case, isolation of biological processes and gene networks affected by treatment is not possible and was not performed.

 Additionally, the discourse on mechanisms interlinking LTP with gene expression merits further elaboration, particularly when contrasting cholinergic, GABAergic, and glutamatergic neurons.

The complex structure of MSA makes a real problem for making final conclusions about the role of cholinergic, GABAergic, and glutamatergic neurons in LTP and gene expression. Therefore, we extended discussion by our thoughts on the existing problems and directions of future studies.

 It is worth noting that in line 318, the mention of the 'epigenetic landscape' lacks clarity regarding its specific reference. If this pertains to a potential explanation for gene expression changes, a more detailed exploration is crucial. Furthermore, citing research that links depolarization to alterations in the epigenetic profiles of various neuronal types would strengthen the manuscript's assertions.

We added to discussion references and short description of epigenetic changes that can be induced by spreading depression.

Reviewer 2 Report

1. I still find the lack of punctuation, please revise.

2. In Line 379, I can't find any information about "stereotaxic coordinates" from reference [33].

  •  

The English still needs to be improved.

Author Response

We would like to thank reviewer for fruitful suggestions. We checked our manuscript for errors and corrected them. All corrections were highlighted.

Reviewer 3 Report

This paper adequately addresses previous revisions, and I feel that the illustrations have been revised to make them easier to see. No further corrections are required. paper adequately addresses previous revisions, and I feel that the illustrations have been revised to make them easier to see. No further corrections are required.This paper adequately addresses previous revisions, and I feel that the illustrations have been revised to make them easier to see. No further corrections are required.

This paper adequately addresses previous revisions, and I feel that the illustrations have been revised to make them easier to see. No further corrections are required.の結果

 This paper adequately addresses previous revisions, and I feel that the illustrations have been revised to make them easier to see. No further corrections are required.

This paper adequately addresses previous revisions, and I feel that the illustrations have been revised to make them easier to see. No further corrections are requiredThis paper adequately addresses previous revisions, and I feel that the illustrations have been revised to make them easier to see. No further corrections are required.

Author Response

(The authors gave the same response as above.)
